

# Sensing and processing whisker deflections in rodents

Thomas F. Burns and Ramesh Rajan

Biomedicine Discovery Institute, Monash University, Melbourne, Victoria, Australia

## ABSTRACT

The classical view of sensory information mainly flowing into barrel cortex at layer IV, moving up for complex feature processing and lateral interactions in layers II and III, then down to layers V and VI for output and corticothalamic feedback is becoming increasingly undermined by new evidence. We review the neurophysiology of sensing and processing whisker deflections, emphasizing the general processing and organisational principles present along the entire sensory pathway—from the site of physical deflection at the whiskers to the encoding of deflections in the barrel cortex. Many of these principles support the classical view. However, we also highlight the growing number of exceptions to these general principles, which complexify the system and which investigators should be mindful of when interpreting their results. We identify gaps in the literature for experimentalists and theorists to investigate, not just to better understand whisker sensation but also to better understand sensory and cortical processing.

## INTRODUCTION

Sensory information is vital for interacting with the world and fulfilling the basic requirements for human survival, socialization, and cooperation. We use combinations of sensory data streams—auditory, visual, touch, etc.—to interpret and form internal maps of the outside world. It is therefore often debilitating for humans to experience minor or major sensory deficits or perturbations in sensory processing pathways, and such debilitations come at broader social and economic costs (*WHO Programme for the Prevention of Deafness and Hearing Impairment, 2001*; *WHO Programme for the Prevention of Deafness and Hearing Impairment, 2010*). For example, partial or complete hearing loss can generate significant stress on familial bonds, general communication, and personal confidence (*Lucas, Katiri & Kitterick, 2018*; *Wood-Jackson & Turnbull, 2004*). Thus, the study of such sensory systems and their deficits is important for human health and wellbeing. Sensory systems are also ideal for basic neuroscience study because sensory information is processed in dedicated, known and highly-structured neural pathways from periphery to cortex and, for the purposes of experimentation, sensory input can be easily manipulated over scales from reductionist through to complex naturalistic stimuli. This allows detailed study of broad questions about how neurons connect and form networks to carry out computations, as well as how different alterations in health states affect these

Corresponding authors
Thomas F. Burns,
t.f.burns@gmail.com
Ramesh Rajan,
ramesh.rajan@monash.edu

networks. The current review focusses on barrel cortex, a highly specialized region of rat cortex for sensing and processing whisker deflections. By understanding this system in detail, it may enable us to generate more a more general understanding of sensory systems and processing.

For bioethical and practical reasons, it is not always preferable or possible to study neural tissues from humans. However, ethical guidelines have been developed for the use of non-human animals (*Blakemore et al., 2012*), and brain development and likeness to human brains in this context has been studied in detail in relevant non-human animals, particularly rodents (*Semple et al., 2012*) which have become popular models for studying brain injury (*Carron, Alwis & Rajan, 2016*) and for basic neuroscience generally (*Reid & Koch, 2012*). Given the ecological niche they occupy, the rodent face whiskers act as important sensing organs to probe the environment and to interact with conspecifics, especially in the low-light nocturnal conditions when rodents are most active. The whiskers are readily manipulated and the relevant parts of somatosensory cortex are easily accessed via surgery; hence, study of whisker-sensation in rodents has been ongoing in basic and sensory neuroscience for many years (*Feldmeyer et al., 1999*; *Jensen & Killackey, 1987*; *Lavzin et al., 2012*; *Phoka et al., 2012*; *White & Rock, 1981*), including to test the effects of deficit or damage on neural function and structure (*Alwis et al., 2012*; *Carron, Alwis & Rajan, 2016*; *Johnstone et al., 2014*).

## RATIONALE AND SURVEY METHODOLOGY

In this review, we outline the basic state of knowledge of rodent whisker-sensation physiology and identify several opportunities for investigators to expand our understanding through filling gaps in the literature. This literature review is therefore useful for junior and senior researchers interested in sensory neuroscience, particularly whisker sensation in rodents.

To find relevant literature for this review, we used the PubMed and Google Scholar literature search engines. We searched for literature including all combinations of the following keywords in their titles or abstracts: "barrel cortex", "whisker", "sensory processing", "sensory pathway", "thalamocortical pathway", "somatopy", "microcircuit", "inhibition", "septa", and "cortical column". Results from these searches were combined between all searches and between PubMed and Google Scholar such that all duplicates were discarded. Only peer-reviewed original research, reviews, and book chapters were included. Results were excluded if the paper/review/book chapter focused on animals which were not mice or rats or focused on aspects of the physiology or biology which was unrelated to whisker sensation.

The main part of our review is structured to match the information flow through the sensory pathways of whisker sensation in rodents, starting from the whisker itself and ending in the barrel cortex. We first outline the general structural elements in the pathways which allow rats to receive information about the world from their whisker movements. Next, we discuss the importance of different neuron subtypes in the microcircuits of the somatosensory cortex, where perception occurs. We then briefly discuss the concept of population coding, especially temporal coding, in the context of barrel cortex.

## Whisker-sensing pathways to barrel cortex

Sensory whiskers, or vibrissae, are composed of a keratin shaft which extends out from a mammal's body. The shaft itself does not contain nerve fibres but its mechanical manipulation activates mechanoreceptors at its base (*Dehnhardt et al., 1999*; *Ebara et al., 2002*; *Kim et al., 2011*; *Marshall et al., 2006*; *Melaragna & Montagna, 1953*; *Rice, Mance & Munger, 1986*; *Stuttgen, Ruter & Schwarz, 2006*). Despite its lack of nervous tissue, the geometry and mechanical characteristics of the whisker itself has significant follow-on effects ascending the sensory pathway. For example, isolated whiskers have been shown to naturally resonate at frequencies ranging from 30–750 Hz (where frequency is proportion to whisker length) and passively damp vibrations or deflections (*Hartmann et al., 2003*; *Neimark et al., 2003*). Sinus pressure near the base of the whisker may further aid in the passive, reflexive, or active modulation of the whisker's mechanical responses and thus the overall sensory transduction (*Neimark et al., 2003*; *Rice, Mance & Munger, 1986*), as originally suggested for shrews (*Yohro, 1977*). The mechanoreceptors which ultimately send this sensory information exist in a wide variety (*Melaragna & Montagna, 1953*) and also exhibit different stimulus feature specificities and firing adaptation rates (*Lichtenstein, Carvell & Simons, 1990*; *Zucker & Welker, 1969*).

In the case of the rat, whiskers are innervated by a purely sensory branch of the ventrolateral trigeminal nerve called the infraorbital nerve. First-order bipolar afferent neurons innervate the whisker base and synapse in the sensory trigeminal nuclei (*Lazarov, 2002*). From here, second-order afferents project to various subcortical nuclei in the brain (*Morton, 2013*) in complex, detailed arrangements (*Castro-Alamancos, 2015*; *Castro-Alamancos, 2002*; *Miyata & Imoto, 2006*) (see Fig. 1 for a summary of these pathways). The main divisions of these afferent pathways to thalamus are the lemniscal, paralemniscal, and extralemniscal. This leads to independent levels of activation of the ventroposterior medial nucleus (VPM) and the medial posterior nucleus (POm), the two major thalamic nuclei which will relay the information to cortex. VPM receives afferents from the lemniscal pathway (in its doromedial sector) and the extralemniscal pathway (in its ventrolateral sector) (*Pierret, Lavallee & Deschenes, 2000*), whereas POm receives afferents only from the paralemniscal pathway. The connection patterns of second-order afferents to VPM and POm achieve two things. First, the system maintains a highly organised and systematic representation of information sent by the individual whiskers (*Land, Buffer & Yaskosky, 1995*; *Pierret, Lavallee & Deschenes, 2000*; *Saporta & Kruger, 1977*). Second, it establishes VPM and POm as distinct, specialised information streams relaying different types of information to cortex (see Fig. 1 for illustration). For example, the lemniscal pathway via VPM may encode spatial information while the paralemniscal pathway via POm encodes temporal information (*Ahlssar, Sosnik & Haldarilu, 2000*; *Yu et al., 2006*). The extralemniscal pathway via VPM may represent a combination of spatial and temporal information (*Yu et al., 2006*), which may or may not make VPM a more general mixture of information, except to say that specific divisions within VPM seem well-defined and differentially innervate the cortex (*Pierret, Lavallee & Deschenes, 2000*). The VPM and POm pathways also show differences in corticothalamic interactions, modulating their own

**Figure 1  Simplified diagrammatic overview of known pathways of input to and interactions with a single barrel column.** Still finer details exist within these structures but are not shown, e.g., barrelletes (found within the brain stem and a sub-area of PrV). Intra-barrel column microcircuits also play an important dynamical role in modulating these pathways and is discussed in the text. Arrowheads indicate direction of projection, "+" symbol indicates relatively strong connections, and lighter colours indicate relatively weak connections (Key: green, excitatory; red, inhibitory; purple, modulatory. Abbreviations same as text.).

responses to activity in cortex (*Diamond, Armstrong-James & Ebner, 1992*) and superior colliculus (*Gharaei et al., 2020*).

The reticular nucleus of the thalamus (NRT) is a common source of inhibition to both VPM and POm. It is activated by corticothalamic feedback and can provide lateral inhibition to neighbouring segments of thalamus representing other, nearby whiskers to inhibit the primary whisker represented by another given segment of thalamus (*Fernandez et al., 2017*; *Landisman et al., 2002*; *Lavallée & Deschênes, 2004*; *Sohal & Huguenard, 2003*; *Varga et al., 2002*). VPM primarily receives inhibition via NRT whereas POm receives additional, intra- and extra-thalamic inhibition (*Bokor et al., 2005*). These additional sources of inhibition normally silence POm activity during regular whisker activation (*Barthó, Freund & Acsády, 2002*; *Lavallée et al., 2005*; *Trageser & Keller, 2004*). The activity of VPM and POm are also susceptible to corticothalamic feedback, especially from cortical layers V and VI (*Bourassa, Pinault & Deschenes, 1995*; *Castro-Alamancos, 2004*; *Castro-Alamancos & Calcagnotto, 1999*; *Deschênes, Veinante & Zhang, 1998*; *Golshani, Liu & Jones, 2001*; *McCormick & Von Krosigk, 1992*; *Sherman & Guillery, 1998*). Corticothalamic fibres sent to POm are thought to the main drivers of POm, and fibres sent to VPM help modulate incoming activity. Corticothalamic fibres originating from cortical layer VI have especially strong synapses (*Hoogland et al., 1991*) and are similar in strength to second-order afferent synapses (*Reichova, 2004*) (while most other corticothalamic synapses are relatively weak). In addition to these cortical feedback mechanisms, VPM is additionally modulated by the brainstem (*Castro-Alamancos, 2015*). In combination, these sources of inhibition, feedback, and modulation (see Fig. 1 for diagrammatic summary) therefore shape the responses of intrinsically excitable thalamocortical cells (*Castro-Alamancos, 2002*; *Steriade, Jones & McCormick, 1997*) in VPM and POm, which receive sensory information from second-order afferents. In summary, this means the cortex receives a signal which is processed by sub-cortical structures and also distributed by those structures to arrive in specific parts of cortex.

## Barrel cortex somatotopy and microcircuitry

In rodents, whisker sensation is represented in the posteromedial barrel subfield (PMBSF) region of somatosensory cortex, commonly known as *barrel cortex* due to specializations in the organization of neurons in the input layer IV as described below. The PMBSF occupies approximately 70% of primary somatosensory cortex and 13% of the cortical surface (*Lee & Erzurumlu, 2005*), a disproportionately large part of cortex relative to the small external physical size of the whiskers when compared to other parts of the rodent body. This indicates the ecological importance of whisker somatosensation in comparison to other tactile inputs. The PMBSF is organised somatopically, meaning each of the major facial whiskers is represented, whisker for whisker, in separate columns of neurons extending from the surface to the white matter. Each of these regions, which receives a dominant *principal whisker* (PW) input, is defined anatomically by the organization of layer IV neurons into 'barrel' like structures with a relatively hollow interior. They are laid out in a grid formation in the PMBSF. The cortical layers above and below a layer IV barrel are often referred to as a 'barrel column' and together represent the cortical column responsible for

processing the sensory input from one whisker. The grid formation in PMBSF consists of arcs (columns) and rows of whisker barrels, each representing the same arcs and rows of whiskers found in the whisker pad on the rat face (and this organisation is maintained through the brainstem, thalamus, and cortex). 'A1' represents the top-most whisker at the nose bridge end, 'A2' the whisker one closer to the nose in the same row, and so on, right through to 'E8', which represents the bottom-most whisker at the nose end (n.b. some species-, individual-, and reporting-specific differences will mean a slightly different end whisker count).

Classically, each whisker barrel receives its primary thalamic input mainly at layer IV. This sensory information is then projected up to layers II and III for further processing (along with other local cortical areas), then down to layers V and VI for final output to more distant cortical areas, such as motor cortex (as well as sending feedback to sub-cortical areas) (*Radnikow, Qi & Feldmeyer, 2015*). Layer IV excitatory cells typically have strong, narrow tuning to single whiskers while cells in supra- and infra-granular layers typically show broader, mixed-strength tuning (indicating tuning to more precise, higher-order features, and possibly common to multiple whiskers, as indicated by generally narrower receptive fields in layer IV compared to other layers (*Brecht, Roth & Sakmann, 2003*; *Brecht & Sakmann, 2002*) and more complex sensory information generally being computed and integrated in cortex in layers other than layer IV (*Bale & Maravall, 2018*; *Lyall et al., 2020*; *O'Herron et al., 2020*)). Neurons across all layers, but particularly infragranular layers, can be tuned to temporal or qualitative features of whisker deflection, e.g., directional sensitivity or initial versus sustained parts of deflection. Such differences are the result of interaction between the increasing number of complex microcircuits being identified (*Feldmeyer, 2012*; *Narayanan et al., 2015*; *Vitali & Jabaudon, 2014*) both within and between layers, and within and between barrels (*Bosman et al., 2011*). For instance, the septa are innervated by a separate thalamic pathway to the barrels (see Fig. 1), and their lateral connections between barrels (*Narayanan et al., 2015*) likely modulate individual barrels' activity. Where (*Feldmeyer, 2012*) and how (*Meyer et al., 2010*) thalamocortical cells connect within the cortical layers is another source of these differences. And while the VPM and POm pathways appear to target cortex in complementary ways, the relevant feedforward (*Cruikshank et al., 2010*; *Lavallée et al., 2005*; *Suzuki & Bekkers, 2012*) and feedback (*Feldmeyer, 2012*; *Kim et al., 2014*) systems within cortex make the subsequent interlaminar interactions all the more complex, thus interesting and important for sensory processing. For these reasons, the classical, simple view of information flowing neatly and wholly from layer IV, up to II/III, then down to V is increasingly being reinforced in general principle while also undermined by long lists of special cases. In the following sub-sections, we will attempt to follow the general principle view in detail, layer-by-layer, and discuss the implications of relevant special cases.

**Layer IV**

Layer IV is the primary input layer from the thalamus and typically has narrow, strong tuning to a single PW. It helps to amplify and further filter the thalamic signal and distribute

its activity to other cortical layers, primarily the supragranular layers (*Cowan & Stricker, 2004*; *Feldmeyer, 2012*; *Staiger et al., 2004*).

In the classical pathway, VPM afferents first synapse onto both excitatory and inhibitory cells in layer IV (*White & Rock, 1979*), with more synapses onto excitatory than inhibitory cells as the ratio of excitatory to inhibitory cells in layer IV is approximately 9:1 (*Lefort et al., 2009*). That said, the vast majority (approximately 85%) of synaptic contacts in layer IV are intracortical (*White & Rock, 1979*), i.e., from other areas of (mostly barrel) cortex. Thalamocortical synapses are only slightly more efficacious than intracortical synapses, with evidence that the relative strength of thalamocortical synapses is due to coincident activation of a number of such inputs rather than significantly stronger synapses (*Jia et al., 2014*). Although, their synapses are also slightly more proximal to somas of layer IV spiny stellate, excitatory cells than are intracortical synapses, which may contribute to the fast lateral inhibition suggested by physiological and functional studies on roughness discrimination (*Hartings & Simons, 1998*; *Pinto, Brumberg & Simons, 2000*; *Shoykhet, Doherty & Simons, 2000*; *Temereanca & Simons, 2003*).

The excitatory cells in layer IV are predominantly spiny stellates, star pyramids, and non-star pyramids (*Brecht & Sakmann, 2002*; *Bruno & Sakmann, 2006*; *Jones, 1975*; *Lübke et al., 2000*; *Schubert et al., 2003*; *Staiger et al., 2014*), all innervated by VPM afferents. There are some morphological and functional differences between them (*Egger, Nevian & Bruno, 2008*; *Staiger et al., 2004*), but they mostly differ in connectivity to other layers and columns (*Cowan & Stricker, 2004*; *Egger, Nevian & Bruno, 2008*; *Lübke et al., 2000*; *Schubert et al., 2003*): spiny stellates axons' project almost exclusively within layer IV and to layer II and III, and very rarely to infragranular layers; star pyramids have dendrites which extend from layer IV into II and III and axons which project to layer II, III, within IV, and to infragranular layers; and non-star pyramids are very similar to star pyramids although can also project to neighbouring columns. Despite these differences, excitatory cells in layer IV mainly target layers II and III in the same column (with an overall connectivity probability of ∼10–15% with layer II/III pyramidal cells; *Feldmeyer et al., 2002*; *Lefort et al., 2009*). Within layer IV, excitatory cells appear to form excitatory clusters of ≤∼10 cells in which the cells are highly interconnected (*Lefort et al., 2009*), making individual cells in these clusters highly efficacious in causing action potentials in other cells of the same cluster (*Feldmeyer et al., 1999*).

Excitatory cell activity in all cortical layers is modulated by interneuron cells located locally (from within the same layer of the same column), translaminarly (from other layers within the same column), and laterally (from layers within other columns). These interneurons come in different and complex morphological and electrical varieties, are present and connected in different proportions and manners throughout cortex, and can have inhibitory or excitatory synapses with other cells (though most are inhibitory) (*DeFelipe et al., 2013*; *Halabisky, 2006*; *Markram et al., 2004*). This diversity can make precise identification of interneuron subtypes difficult under experimental conditions, and in many reports of microcircuits, only some features of the interneurons are known. However, a common technique to identify them takes advantage of their differential expressions of calcium-binding proteins (CBPs), neuropeptides, and other molecular

markers (*Druckmann et al., 2013*; *Kubota et al., 2016*; *Markram et al., 2004*) (see Table 1). In the case of layer IV interneurons, VPM afferents activate cells positive for parvalbumin (PV$^+$), somatostatin (SOM$^+$), and 5-HT$_{3A}$ receptors (5-HT$_{3A}$R$^+$) (*Beierlein, Gibson & Connors, 2003*; *Lee et al., 2010*; *Porter, Johnson & Agmon, 2001*). PV$^+$ interneurons are also driven by layer IV excitatory cells (*Bosman et al., 2011*; *Koelbl et al., 2015*) and layer VI corticothalamic pyramidal cells (*Kim et al., 2014*). Together, these excitatory connections onto inhibitory cells can be considered *feedforward inhibition*, as they drive inhibitory interneurons' activity forward onto other cells. *Feedback inhibition* occurs when inhibitory cells synapse back onto excitatory cells, typically releasing gamma-aminobutyric acid (GABA). PV$^+$, SOM$^+$, and 5-HT$_{3A}$R$^+$ layer IV interneurons cause feedback inhibition on layer IV excitatory cells (*Chittajallu, Pelkey & McBain, 2013*; *Koelbl et al., 2015*; *Xu et al., 2013*) and SOM$^+$ interneurons cause *disinhibition* (inhibiting other inhibitory cells, thus reducing their inhibition onto excitatory cells) on PV$^+$ interneurons (*Xu et al., 2013*).

Functionally, PV$^+$ interneurons appear to be mostly fast-spiking (FS) and can produce very high, non-adapting firing rates (>100 Hz). They synapse almost exclusively onto excitatory cells in layer IV and are likely to be basket cells (BCs; see Table 1) which typically possess a dense axonal plexus that projects within a small area (*Koelbl et al., 2015*; *Porter, Johnson & Agmon, 2001*). These PV$^+$-FS cells have very short latencies to cortical activation (0.6 ms), high release probabilities, and make an average of 3.5 synapses onto excitatory cell dendrites at proximal and distal locations (*Koelbl et al., 2015*). It has therefore been suggested (*Radnikow, Qi & Feldmeyer, 2015*) that as these PV$^+$-FS cells are rapidly recruited by thalamocortical afferents and further driven by local excitatory cells, they may act to quickly 'reset' layer IV excitation and increase temporal resolution in that layer. Relatively FS (70–150 Hz), adaptive firing from SOM$^+$ cells likely provides the required disinhibitry control of PV$^+$-FS cells (*Ma et al., 2006*); synapses from SOM$^+$ to PV$^+$- FS cells are much stronger than those from SOM$^+$ to excitatory cells within layer IV (*Xu et al., 2013*). However, a different subtype of SOM$^+$ interneuron in layer IV are likely to be the Martinotti cells which are identifiable by axons projecting to layer I (*Ma et al., 2006*) and provide widespread cortical dampening to pyramidal neurons (*Silberberg & Markram, 2007*) (Table 1). Then, 5-HT$_{3A}$R$^+$ interneurons—which appear in comparatively low numbers in this layer—show long firing latencies and result in slow inhibition on excitatory cells within layer IV (*Chittajallu, Pelkey & McBain, 2013*; *Lee et al., 2010*; *Rudy et al., 2011*), acting weakly but surely against PV$^+$-FS cells' temporal sharpening. This could counteract excitation-inhibition imbalances or provide the wider temporal integration necessary for long-term neuroplasticity (*Radnikow, Qi & Feldmeyer, 2015*).

## Layer II and III

The main layer IV excitatory output is to layers II and III, where it combines with additional VPM input to layer III pyramidal cells (*Arnold, Li & Waters, 2001*; *Jensen & Killackey, 1987*; *Meyer et al., 2010*; *Oberlaender et al., 2012*) and POm input to apical tufts of layer II pyramidal cells (*Ohno et al., 2012*; *Radnikow, Qi & Feldmeyer, 2015*). Together, layers II and III act as the first and major integrative processing cortical layers. Pyramidal cells in

# PeerJ

**Table 1** Summary of interneuron sub-types mentioned in text (abbreviations same as text).

| Name | Basic description | Identifying characteristics (*Druckmann et al., 2013*; *Markram et al., 2015*; *Markram et al., 2004*; *Radnikow, Qi & Feldmeyer, 2015*) |
|---|---|---|
| Spiny stellate cell (SSC) | Filters and relays thalamic excitation from layer IV to layer II/III (*Schubert et al., 2003*). | Spiny morphology. |
| Large basket cell (LBC) | Most common basket cell found in cortex (*Markram et al., 2004*). Inhibits many pyramidal cells across barrels at or near the soma (Wang, 2002). | FS, non-accommodating (N-Ac), non-adapting (N-Ad). PV (+++), CB (++), NPY (+), CR (+), VIP (+), SOM (-), CCK (+). |
| Small basket cell (SBC) | Least common basket cell found in cortex (*Markram et al., 2004*). Inhibits few pyramidal cells, usually within a single layer and column, at or near the soma (Wang, 2002). | FS, N-Ac, N-Ad. PV (-), CB (++), NPY (+), CR (-), VIP (+++), SOM (++), CCK (+). |
| Nest basket cell (NBC) | Second-most common basket cell found in cortex (*Markram et al., 2004*). Inhibits few pyramidal cells within a barrel at or near the soma (Wang, 2002). | FS, N-Ac, N-Ad. PV (+++), CB (++), NPY (+), CR (++), VIP (+), SOM (-), CCK (+). |
| Chandelier cell (ChC) | Inhibits the initial segment of pyramidal neurons and found in layers II to VI (*Borden, 1996*; *Hardwick et al., 2005*; *Markram et al., 2004*; *Taniguchi et al., 2013*; *Woodruff & Yuste, 2008*) to control excessive excitation (*Zhu et al., 2004*), although some connections could themselves be excitatory (*Szabadics et al., 2006*). | FS or late spiking (LS), N-Ad. PV (+) and/or CB (+), GABA transporter 1 (GAT-1) (+), SOM (-). |
| Neurogliaform cell (NGFC) | Inhibits dendrites of pyramidal neurons (*Markram et al., 2004*), especially in instances of persistent excitation (*Overstreet-Wadiche & McBain, 2015*; *Suzuki & Bekkers, 2012*). | LS. PV (-), CB (-), NPY (+), reelin (+), COUP transcription factor 2 (+). |
| Double bouquet cell (DBoC) | Inhibits basal dendrites and somas of pyramidal neurons (*Markram et al., 2004*), typically extending its dendrites vertically, across multiple layers (*Kawaguchi & Kubota, 1996*; *Krimer, 2005; Somogyi; Cowey, 1984*). | Irregular spiking (IS) or regular non-pyramidal (RSNP) firing (adapting). PV (-), CB (-), NPY (-), CR (++), VIP (+++), SOM (++). |
| Bitufted cell (BTC) | Inhibits distal dendrites of pyramidal neurons (*Markram et al., 2004*), often spanning its dendrites across the entire cortical column (*Kaiser et al., 2001*; *Peters & Harriman, 1988*; *Tamás et al., 1998*). | RSNP and BSNP, adapting. PV (-), CB (++), NPY (+), CR (++), VIP (+), SOM (++). |
| Bipolar cell (BPC) | Extends narrow bipolar or bitufted dendrites vertically within the column. Inhibits the basal dendrites of relatively few pyramidal neurons (*Markram et al., 2004*). | IS, LS, or RSNP (adapting). PV (-), CB (-), NPY (-), CR (++), VIP (+++), SOM (++). |
| Multipolar bursting cell (MPBC) | Extends densely within layer II, with some collaterals to layer V. Inhibits the basal dendrites of local pyramidal neurons (*Blatow et al., 2003*; *Caputi et al., 2009*). | Burst firing. CR (+). |
| Martinotti cell (MC) | Inhibit distal dendrites of pyramidal neurons (*Markram et al., 2004*), especially the apical tuft regions in layer I (of deeper pyramidal neurons). | RSNP or burst-spiking non-pyramidal (BSNP). PV (-), CB (++), NPY (++), CR (-), VIP (-), SOM (+++). |
| Single bouquet cell (SBoC) | Inhibits interneurons in supragranular layers, indirectly disinhibiting layer V pyramidal neurons (*Jiang et al., 2013*; *Larkum, 2013*; *Lee et al., 2014*). | Varied spiking patterns. Typically VIP (+). |
| Elongated neurogliaform cell (ENGFC) | Inhibits distal dendrites present in layer I, typically the apical tufts, of layer II, III, and V pyramidal neurons (*Jiang et al., 2013*; *Larkum, 2013*; *Lee et al., 2014*). | LS and varied spiking patterns. Typically NPY (+) and reelin (+). |

**Table 1** (*continued*)

| Name | Basic description | Identifying characteristics (*Druckmann et al., 2013*; *Markram et al., 2015*; *Markram et al., 2004*; *Radnikow, Qi & Feldmeyer, 2015*) |
|---|---|---|
| Cajal-Retzius cell | Important for establishing intracortical and cortico-thalamic connections during development (*Del Río et al., 1997*; *Hevner et al., 2003*; *Imamoto et al., 1994*; *Meyer et al., 1999*; *Soriano & Del Río, 2005*), although some may survive into adulthood (Meyer et al., 1999; Soriano & Del Río, 2005). | Glutamergic. Typically reelin (+). |

layer II and III typically project their axons over several barrel columns in layers II, III, and V, and to secondary somatosensory and motor cortices (*Aronoff et al., 2010*; *Feldmeyer, Lübke & Sakmann, 2006*). However, layer II pyramidal cells near the border of layers I and II have highly lateralized apical dendrites, and a small subset of layer III pyramidal cells restrict their projections to mostly within one barrel (*Bruno et al., 2009*; *Larsen & Callaway, 2006*). Within layer II and III, pyramidal neurons form excitatory connections to one another with a probability of ∼10–20%, as layer IV excitatory cells connect to layer II and III pyramidal cells (*Feldmeyer, Lübke & Sakmann, 2006*; *Holmgren, Harkany & Zilberter, 2003*). These intralayer connections between layer II and III pyramidal cells are typically on the order of ∼3 synaptic connections per neuron (to mostly basal dendrites) (*Feldmeyer, Lübke & Sakmann, 2006*; *Sarid et al., 2015*), however the strength of these connections depends on sensory experience (*Cheetham et al., 2007*). Axons from layer II and III pyramidal neurons also project to layer V pyramidal neurons, typically forming weak synapses on basal dendrites (*Petreanu et al., 2009*; *Reyes & Sakmann, 1999*; *Schubert et al., 2006*), and these connection patterns may 'bind' perceptual features in subnetworks of layer V pyramidal cells through learning rules such as spike timing-dependent synaptic plasticity (*Kampa, Letzkus & Stuart, 2006*) and to generate combinations of such features for output to other cortical areas (see section 'Layer V'). .

As in layer IV, the output of layer II and III pyramidal neurons is shaped by many interneurons, particularly in layer II where ∼17% of cells are interneurons, whereas interneurons make up only ∼9% in layer III and ∼8–9% in layer IV (*Meyer et al., 2011*). All major histological classes of interneurons are represented in layers II and III (*Gentet, 2012*) but approximately half are 5-HT$_{3A}$R$^+$, meaning they can be driven by serotonergic neurons (*Rudy et al., 2011*). Layer II and III interneurons are mainly driven by layer IV excitatory cells (*Helmstaedter et al., 2008*), causing feedforward inhibition, but layer II and III pyramidal neurons also activate feedforward inhibition circuits by synapsing with some FS (possible BC) layer II and III interneurons (*Avermann et al., 2012*; *Holmgren, Harkany & Zilberter, 2003*). A wide variety of interneurons, each with unique intrinsic properties and functions, are likely to exist in layer II and III, including BCs, Martinotti cells (MCs), chandelier cells (ChCs), neurogliaform cells (NGFCs), double bouquet cells (DBCs), bitufted cells (BTCs), and bipolar cells (BPCs) (*DeFelipe et al., 2013*; *Jiang et al., 2013*; *Lee et al., 2014*; *Markram et al., 2004*) (see Table 1 for basic descriptions of these cells and their identifying characteristics). ChCs target axon initial segments of layer II and III pyramidal neurons (where they can be uniquely excitatory (*Szabadics et al., 2006*)), while BCs, DBCs,

and some NGFCs synapse onto basal dendrites of local pyramidal neurons; BPCs target proximal apical dendrites; and MCs and BTCs synapse on apical tufts and the middle portion of the apical dendrites. On average, most of these interneurons will synapse onto three to six pyramidal neurons in layer II and III and make similar kinds and numbers of connections with layer V pyramidal neurons. These interneuronal connections onto layer II and III pyramidal cells, in combination with the intricate excitatory connection patterns from layer IV and thalamus, allows cells to be finely tuned to complex, higher-order features of sensory input (*Bale & Maravall, 2018*; *Lyall et al., 2020*; *O'Herron et al., 2020*).

In addition to these local interactions, there are also some interesting BC-involving microcircuits in which BCs are innervated by long-range vibrassal motor cortex (vM1) axons which synapse onto BCs' apical dendrite extensions in layer I of barrel cortex. These BCs cause a strong disinhibition of SOM$^+$ interneurons in layer II and III, and thus activity from vM1 projections can increase the excitability of layer II, III, and V pyramidal neurons, as seen during whisking behaviour in vivo (*Lee et al., 2013*; *Xu et al., 2013*). Layer IV excitatory neurons in secondary somatosensory cortex have also been shown to drive an important long-range feedback pathway to barrel cortex, affecting orientation tuning within barrel cortex (*Minamisawa et al., 2018*).

Highly peculiar layer II and III inhibitory microcircuits involving PV$^+$ and calretinin-positive (CR$^+$) interneurons have also been observed (*Blatow et al., 2003*; *Caputi et al., 2009*). These PV$^+$ cells are called multipolar bursting cells (MPBCs) as they show burst rather than FS firing when depolarised and project densely within layer II, with some collaterals to layer V. The CR$^+$ cells are BPCs and multipolar cells (MPCs)—BPCs project narrowly down to layer V and, like MPBCs, have a high-frequency burst upon initial depolarisation, whereas MPCs' axons project laterally within layer II and III only. These peculiar circuits are driven and modulated by layer II and III pyramidal cells, with MPBCs receiving extra inputs from layer IV excitatory cells.

**Layer V**

Layer V receives excitation and inhibition from all overlying layers and, combined with excitatory input from VPM and POm, likely integrates the processing of the column as a whole before sending its processed output to downstream areas. The substantial thalamic input, particularly from VPM (*Constantinople & Bruno, 2013*), also challenges the conventional view of sensory information mainly arriving in cortex at layer IV. Layer V pyramidal neurons receive innervation from supragranular and granular excitatory cells, as well as other layer V pyramidal neurons. Of these supragranular and granular cells, three morphologies are distinguishable: slender-tufted (found mostly in upper layer V), thick-tufted (found mostly in lower layer V), and untufted (found throughout layer V, though in relatively low numbers) (*Feldmeyer, Lübke & Sakmann, 2006*; *Larsen & Callaway, 2006*). Slender-tufted pyramidal neurons receive thalamic input from POm and project dense axons extensively within supragranular layers across the entire ipsilateral barrel cortex, ipsilateral vM1, and to contralateral barrel cortex, making them the primary output cells in layer V (*Larsen, Wickersham & Callaway, 2007*; *Oberlaender et al., 2011*). Thick-tufted pyramidal neurons receive thalamic input mostly from VPM and make most

of their synapses to other layer V pyramidal neurons and subcortical areas, thus providing local and subcortical feedback (*Larsen, Wickersham & Callaway, 2007*; *Veinante, Lavallée & Deschênes, 2000*). Untufted pyramidal neurons receive a mix of VPM and POm thalamic input depending on their depth within layer V and project extensively to layer III and to the contralateral barrel cortex, making them important for intracolumnar feedback and inter-hemispheric coordination of sensory outputs (*Larsen & Callaway, 2006*; *Le Bé et al., 2007*).

As in other layers of cortex, the activity of pyramidal neurons in layer V receives inhibitory modulation—from other layers, especially layer II and III interneurons as discussed above, as well as local layer V inhibition. Local Inhibition comes from $PV^+$, FS and $SOM^+$ cells driven by VPM (*Tan et al., 2008*), FS cells driven by upper layer VI pyramidal cells (*Kim et al., 2014*), and local MCs (*Berger et al., 2010*; *Silberberg & Markram, 2007*). Local $PV^+$, FS and MCs generally behave as in other cortical layers but $SOM^+$ cells synapse onto dendrites of layer IV spiny stellate cells instead of apical dendrites of local pyramidal cells as in other layers (*Tan et al., 2008*). Because of this, their delayed facilitation response effectively adds a late-onset inhibitory input to layer IV excitation during long periods of ongoing thalamocortical input and so could be important for excitation-inhibition balance at longer-time scales.

## Layer I

Layer I, also known as the molecular layer, contains few neuron cell bodies and many glia. It receives input from thalamic matrix cells and acts as a medium through which feedback and transmission from ipsilateral and contralateral cortical areas can communicate (*Jiang et al., 2013*; *Lee et al., 2014*; *Rubio-Garrido et al., 2009*; *Wozny & Williams, 2011*). Except for Cajal-Retzius cells, which are important during neurodevelopment (*Hevner et al., 2003*; *Imamoto et al., 1994*), the mature layer I almost exclusively contains GABAergic inhibitory neurons expressing $5\text{-}HT_{3A}R^+$ and $SOM^+$ (*Rudy et al., 2011*; *Xu et al., 2012*), which are predominately driven by layer II/III pyramidal cells from the same column (*Wozny & Williams, 2011*). These layer I interneurons have some functional spiking differences (*Wozny & Williams, 2011*) and have recently been described as possessing two distinct morphologies, each being involved in two distinct microcircuits (*Jiang et al., 2013*; *Larkum, 2013*; *Lee et al., 2014*): (1) single bouquet cells (SBoCs) establish local, unidirectional inhibition to layer I interneurons and most inhibitory and pyramidal cells in layer II and III; and (2) elongated neurogliaform cells (ENGFCs) establish broad, reciprocal inhibition (directly and via gap junctions) to layer II and III MCs, NGFCs, and BTCs, as well direct inhibition to layer II, III, and V pyramidal neurons. Therefore, SBoCs exert an indirect, disinhibitory effect on layer V pyramidal cells whereas ENGFCs exert direct and indirect inhibition on layer V pyramidal cells (thus stipulated as a yin and yang system of inhibitory control for layer V (*Larkum, 2013*)).

## Layer VI

Relative to other cortical layers, much less is known about layer VI (*Thomson, 2010*). Its involvement in thalamocortical feedback is well-established (*Lam & Sherman, 2010*;

*Mercer et al., 2005*; *Perrenoud et al., 2013*; *West et al., 2006*) but it also makes some unique and varied projections to other layers and acts as a supplementary output layer (*Kumar & Ohana, 2008*; *Lefort et al., 2009*; *Mercer et al., 2005*; *Pichon et al., 2012*; *Zhang & Deschênes, 1997*). In upper layer VI, pyramidal cells receive input from VPM and POm, and typically project either back into the thalamus, providing corticothalamic feedback, or project intracortically. Intracortical projections are made to other layers of the same cortical column, within layer VI itself, or to long-range cortical areas outside of barrel cortex. Pyramidal cells in lower layer VI are much more heterogeneous and typically synapse within layer VI, however longer-range connections to layer I, layer II, and the thalamus are also present (*Clancy & Cauller, 1999*; *Killackey & Sherman, 2003*; *Marx & Feldmeyer, 2013*). Interneurons in layer VI are significantly understudied, but are likely driven by thalamic nuclei and local pyramidal cells and appear to be involved in local and translaminar inhibition (*Bortone, Olsen & Scanziani, 2014*; *Cruikshank et al., 2010*; *Perrenoud et al., 2013*; *West et al., 2006*). Functionally, this makes layer VI highly important for thalamic-cortex interaction, supplementary corticocortical output, and specialised inhibition within a barrel.

Throughout this discussion of barrel cortex microcircuits, some dendritic processing effects have been implied via mention of where presynaptic neurons synapse on postsynaptic cells (soma, perisoma, basal or distal dendrites, etc.). It is important to explicitly note that dendroarchitecture plays an important role in synapse (and, thus, postsynaptic cell) function (*Araya, 2014*; *Bar-Ilan, Gidon & Segev, 2013*; *Jia et al., 2014*; *Kurotani et al., 2008*; *Lavzin et al., 2012*; *Schoonover et al., 2014*; *Stuart, 2012*; *Varga et al., 2002*). For example, GABA$_B$ receptors work by different biochemical mechanisms in the soma than in dendrites; mostly this means that they have the same resulting effect on postsynaptic firing (*Breton & Stuart, 2012*) however exceptions can and do arise due to these differences (*Stuart, 2012*). Complete exploration of such exceptions is beyond the scope of this review but underscores a caveat of dual intracellular somatic recordings (a technique which some studies discussed so far have used): if one is attempting to establish connection probability from presynaptic neurons to a postsynaptic cell, distal dendritic synapses may be so attenuated or filtered that they fail to register at the soma (where the experimenter is often recording from in such studies). However, despite their distal locations, there may be strategies such presynaptic neurons use to boost their signal in vivo (such as synchronous firing with other presynaptic neurons). Thus, in the absence of observing such strategies, studies may significantly underestimate the probability of these presynaptic connections (*Radnikow, Qi & Feldmeyer, 2015*).

In this section on barrel cortex we have also made reference to multiple inhibitory neurons sub-types and their functions (also summarised in Table 1). As a general synthesis of their numerous and complex interactions, both with excitatory neurons and other inhibitory neurons, we notice three general trends: (1) in layers where there are more inhibitory neurons, there is generally more sophisticated or higher level sensory features encoded (layer IV has very few inhibitory neurons compared to layers II and III, for example (*Lefort et al., 2009*)); (2) inhibitory neurons can act as powerful network control mechanism (we can perhaps see this most clearly in the yin and yang dichotomy of layer

I's inhibitory function (*Larkum, 2013*), but this also evident in long-range connections from other cortical areas (*Lee et al., 2013*)); and (3) healthy barrel cortex maintains its overall excitatory–inhibitory balance in spite of the diversity of inhibitory neurons in type, number, and proportion in comparison to excitatory cells across different layers, likely due to layer-specific circuitry and connection probabilities (with supports evidence from brain injury models, showing layer-specific inhibitory and excitatory–inhibitory balance effects (*Johnstone et al., 2013*)).

Nevertheless, much is known about barrel cortex and its microcircuits. The sum of interactions between cortical microcircuits across and within layers ultimately leads to corticocortical output (*Aronoff et al., 2010*), which transmits essential sensory information upstream to higher sensory, motor, and associative cortical areas.

## Encoding of whisker deflections

What we have discussed—whisker-sensing pathways to barrel cortex, and the barrel cortex's somatotopy and microcircuitry—ultimately serves to generate the neural encoding of whisker movements in the brain. Such encoding relies not on individual neurons alone, but rather on populations on neurons spread across the cortical column. While populations within the same barrel tend to do have redundancy (*Panzeri et al., 2003*; *Petersen, Panzeri & Diamond, 2001*), there exist different populations within each barrel, each of which encode complementary features of whisker movements in concert with one another to represent a wide diversity and complexity sensory information (*Adibi et al., 2014*; *Campagner et al., 2018*). For example, different populations can encode whisker deflection velocity, amplitude, or angle of deflection. Such populations can also place different emphases on certain quantitative metrics of their coding system, e.g., neurons encoding stimulus location appear to place an emphasis on the timing of individual spikes, especially of the first spike after stimulus onset (*Petersen, Panzeri & Diamond, 2001*).

Within populations, nearby neurons can fire in pairs during both spontaneous and evoked activity (*Maravall et al., 2007*). This redundancy allows sampling pairs across the population to provide greater accuracy for determining stimulus onset times (*Maravall et al., 2007*), which as mentioned is a vital and sensitive metric for encoding stimulus location (*Petersen, Panzeri & Diamond, 2001*) and many other complex stimulus features of whisker movement (*Adibi et al., 2014*; *Campagner et al., 2018*). However, such representations can and do change over time due to adaptation and sensory experience (*Adibi et al., 2014*; *Maravall et al., 2007*). These encoding mechanisms are important for the healthy function not only of barrel cortex but may also be important for upstream areas which barrel cortex projects to, such as those responsible for functions such as cognition and motor coordination (*Alwis et al., 2012*; *Carron, Alwis & Rajan, 2016*; *Johnstone et al., 2014*). Indeed, recent work has shown animal behaviour selectively transmits information from specific sub-populations of cells within barrel cortex to upstream areas (*Chen et al., 2013*) and help to generate coordination between these higher areas in goal-directed motor tasks (*Chen et al., 2016*).

## CONCLUSIONS

In this review we have discussed the general principle of sensory information being first received at the whisker, then being transmitted and modified via sub-cortical pathways. This information then flows into layer IV of barrel cortex, up to layers II and III for processing and integration, then down to layers V and VI for final processing before output to other areas. Although we emphasized these general processing and organizational principles, we also highlighted many exceptions which complexify the system, e.g., all cortical layers receive some level of input from the thalamus, although in very different quantities and from distinct thalamocortical pathways, or that some neurons within layers II and III form highly connected intra-cortical circuits instead of connecting laterally to neighbouring barrel columns.

Such expectations provide many opportunities to fill gaps in our knowledge of the whisker-sensing system, for example: What are the characteristics of thalamic innervation of the septa? How does such innervation influence activity in adjacent barrels or vice-versa? For example, do lateral interneuronal circuits in layers II and III form functional microcircuits within the septa? Do such circuits properly 'belong' to any one barrel? Do thalamocortical feedback circuits in layer VI—especially those involving interneurons— interact with these supragranular circuits? What functional differences arise in cortex due to modification of VPM or POm input? How are the VPM or POm pathways modulated by thalamocortical interaction and lateral thalamic interactions via NRT? What are the short- and long-term functional implications of feedback arriving to barrel cortex via layer I or indirect modulation of thalamic input (from BFR, APT, or ZI)? For example, do vM1 projections to barrel cortex (which increase excitability of pyramidal neurons in layers II, III, and V) participate in sensory learning or expectation behaviours? Could short-term adaptation in such circuits further explain certain whisking behaviours or strategies?

We believe these and many other questions are worthy of investigation, and that their answers will be relevant to our understanding of other sensory systems and brain processing generally. Many of these questions likely call for close collaboration between experimentalists from different technical backgrounds (electrophysiology, functional imaging, behaviour, immunohistochemistry, anatomy, etc.), as well as computational scientists, engineers, and theorists. Pursuing an even more detailed understanding of how whisker sensation in rodents works is therefore not just good for our scientific understanding, but also for promoting and fostering scientific collaboration across different disciplines.

### Funding
The authors received no funding for this work.

### Competing Interests
The authors declare there are no competing interests.
## Author Contributions

- Thomas F. Burns analyzed the data, prepared figures and/or tables, authored or reviewed drafts of the paper, and approved the final draft.
- Ramesh Rajan analyzed the data, authored or reviewed drafts of the paper, and approved the final draft.

## Data Availability

   This paper did not generate original data as it is a literature review.

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
