# Peer review of "Sensing and processing whisker deflections in rodents"

_PeerJ, doi:10.7717/peerj.10730_

## Round 0.1 · original submission · Major Revisions

· Academic Editor

Major Revisions

As you will see, both reviewers have asked to deepen the content of the article and have provided some examples of where this is needed. I hope you will be able to look at the reviewers' comments so as to address their concerns before submitting a revised manuscript.

·

Basic reporting

The paper is easy to read.

The literature is not comprehensive which is an issue for a review article.

The article is structured appropriately.

Experimental design

This is a review article. If this was meant as a comprehensive review the authors seem to have missed some important papers.

No ethical or methodological issues.

Validity of the findings

not applicable

Additional comments

Burns and Rajan provide a review of the circuitry in the barrel cortex which receives inputs from the contralateral large mystacial vibrissae.
Overall the basic anatomy is well represented, but how this anatomy results in observed physiology, behavior, and responsiveness is missing. The review was a bit on the cursory side and has not broken much new ground.
Major points.
1. The introduction argues for the importance of studying sensory systems in isolation do to its societal importance. Only general examples are provided, a specific case where a sensory modality is lost and how that leads to a problem is needed.
2. When describing the non-cortical pathway of the whisker-to-barrel pathway some important details are wrong (whisker afferents do not synapse within the trigeminal ganglia) or glossed over (the organization of the ‘barrelletes’ within the brainstem and one sub area of PrV (caudalis) is omitted from Figure 1).
3. There is some processing of the signal at the whisker (resonance, adaptive properties), follicle (transduction impacted by many things including the blood sinus), brainstem (there are local inter neurons and the neurons show adaptation) that ultimately impact higher order structures that need to be considered.
4. VPm inputs also target infragranular layers which is not mentioned (Constantinople and Bruno 2013 among others)
5. Feedback connections from S2 and M1 play important roles in S1 layers 2/3 and the impacted circuits are omitted (see Lee et al 2013)
6. How the location of VPm inputs onto spiny stellate (on spines/distal dendrites) versus onto soma and proximal dendrites of FS cells explains cross(adjacent) whisker inhibition would be a good way to link circuits to physiology and then function (roughness discrimination) work of Simons and colleagues.
7. The discussion of interneurons is a bit idiosyncratic and no attempt is made to synthesize, although table 1 is well done.
8. Figure 1 should include arrows to indicate the direction of the connections.
9. Recent work linking specific phenotypes of neurons to specific circuits related to whisker behavior is omitted (see Chen et al 2013, Chen et al 2016)
Minor comment
1. The abstract just provides in essence a table of contents and does not provide any insight as to what the paper will be on
2. Some of the citations within the paper include first names and these should be edited.

Reviewer 2 ·

Basic reporting

In reading this review, I believe that some parts are well-described while others are very superficial. In general, review of the microcircuitry within S1 by layer does contain a useful synthesis of previous research. However, the review fails to place this in a functional context. If that is beyond the scope of this review, then that needs to be clearly motivated in the introduction.

Motivation: I see two main motivations described in the introduction: Line 37: “how neurons connect and form networks to carry out computations” and Lines 39-40 “understanding the brain in health and in injury with a view to creating the basic neuroscience knowledge base for improving human health”. Neither of these topics is fully (or even partially for the second topic) developed in the review.

Based on the introduction I was expecting to see much more detail about how the circuitry enables specific computations and transformations. This was severely lacking, and where attempted, was done so without support (as detailed below).

The description of trigeminal structure was lacking. Historically there are descriptions of lemniscal, paralemniscal and extralemniscal pathways originating from brainstem. What kind of information is processed in each? This is critical to understanding the “computations” of this system.

Some aspects of figure 1 are confusing. Does the VPM core send inhibitory projections to NRT? VPM thalamic input to cortex doesn’t match the canonical pattern, is that on purpose? Cross-excitatory connections between NRT? Excitatory projection from NRT to cortex? Lack of lateral connections in LII/III?

Experimental design

No comment

Validity of the findings

My main concern with the validity of the review is that the authors make numerous bold statements about function, without any support. This includes:

97-98 “VPM and POm as distinct, specialised information streams relaying different types of information to cortex” What types of information? What is the evidence?

122-124 “This results in the cortex, specifically barrel cortex, receiving a partially filtered sensory signal and not the raw sensory signal itself.” How so filtered? What is the evidence?

153-154 “infra-granular layers typically show broader, mixed-strength tuning (indicating tuning to more precise, higher-order features, and possibly common to multiple whiskers” Is there any evidence for this?

273-274 “these connection patterns may ‘bind’ perceptual features in subnetworks of layer V pyramidal cells” Is there any evidence for this?

296-298 “intricate excitatory connection patterns from layer IV and thalamus, allows cells to be finely tuned to complex, higher-order features of sensory input” Is there any evidence for this?

431-433 “These encoding mechanisms are important for the healthy function not only of barrel cortex but also of the upstream areas which barrel cortex projects to, such as those responsible for functions such as cognition and motor coordination.” Is there any evidence for this?

---

## Round 0.2 · accepted · Accept

· Academic Editor

Accept

Please make sure that you make the formatting corrections that have been identified by the reviewer.

·

Basic reporting

The revised manuscript addresses my initial critiques, a little more care to remove the first name initials (e.g. E.L. White....) from some of the citations needs to be done - that did not impact content or readability.

Experimental design

Not applicable - review article

Validity of the findings

the review is logical and following revision highlights all the relevant points.

Additional comments

No further concerns - very minor revision on the citations and then acceptable